# Knowledge, attitudes, and practices among Indonesian urban communities regarding HPV infection, cervical cancer, and HPV vaccination

**Hariyono Winarto**[1]☉*, **Muhammad Habiburrahman**[2]☉, **Maya Dorothea**[2]‡,
**Andrew Wijaya**[3]‡, **Kartiwa Hadi Nuryanto**[1]‡, **Fitriyadi Kusuma**[1]‡, **Tofan Widya Utami**[1]‡,
**Tricia Dewi Anggraeni**[1]‡

**1** Faculty of Medicine Universitas Indonesia, Division of Gynecologic Oncology, Department of Obstetric and Gynaecology, Dr. Cipto Mangunkusumo Hospital, Greater Jakarta, DKI Jakarta, Indonesia, **2** Faculty of Medicine Universitas Indonesia, Dr. Cipto Mangunkusumo Hospital, Greater Jakarta, DKI Jakarta, Indonesia, **3** Faculty of Medicine Universitas Indonesia, Department of Obstetric and Gynaecology, Dr. Cipto Mangunkusumo Hospital, Greater Jakarta, DKI Jakarta, Indonesia

☉ These authors contributed equally to this work.
‡ These authors also contributed equally to this work.
* hariyono.winarto@ui.ac.id

**Data Availability Statement:** All relevant data are available on Dryad (doi:10.5061/dryad.ns1rn8pv4).

## Abstract

### Background

Few studies explored Indonesian understanding of cervical cancer (CC) and the human papillomavirus (HPV) vaccination. We aimed to investigate the association between knowledge, attitudes, and practices (KAP) and socio-demographical influences related to HPV, CC, and vaccination among Indonesian urban citizens.

### Methods

We conducted an online survey during March 2020-August 2021 using the Snowball sampling technique. The socio-demographic characteristic and KAP responses were collected via Google Forms from 400 respondents in Jakarta. The knowledge and attitudes were divided into HPV and CC (aspect 1) and HPV vaccination (aspect 2). Correlation between KAP scores was performed using Spearman's test, and multiple logistic regression analyses were conducted to determine KAP predictors.

### Results

Indonesian urban citizens in Jakarta were found to have poor knowledge in individual aspects of the inquiry but moderate knowledge overall, good attitude in inquiry both in each aspect and overall, and unsatisfying practices. Overall, in the general population, men, and women respectively: 50.8%, 32.4%, and 53.6% had good knowledge; 82.0%, 75.2%, and 84.4% expressed positive attitude; and 30.3%, 15.2%, and 35.6% applied favorable practice regarding questions inquired. Knowledge was weakly correlated towards attitude (ρ =

**Funding:** The authors received no specific funding for this work.

**Competing interests:** The authors have declared that no competing interests exist.

0.385) but moderately correlated with practice ($\rho = 0.485$); attitude was moderately correlated with practice ($\rho = 0.577$), all results: $p < 0.001$. Significant odds ratio (OR) for predictors to good knowledge were female sex (OR = 2.99), higher education (OR = 2.91), and higher mother's education (OR = 2.15). Factors related to positive attitudes were higher mother's education (OR = 4.13), younger age (OR = 1.86), and better results in the knowledge inquiries (OR = 2.96). Factors that suggested better practices were female sex (OR = 2.33), being employed (OR = 1.68), excellent knowledge scores (OR = 4.56), and positive attitudes expressions (OR = 8.05). Having done one vaccination dose and intention to receive vaccines were significantly influenced by good KAP.

## Conclusions

KAP had inter-association to successful CC and HPV prevention programs, and socio-demographical characteristics are critical to influencing better KAP.

## Introduction

Cervical cancer (CC) is the most prevailing gynecological cancer afflicting women aged 15 to 44 [1]. In 2020, CC was the fourth most common cancer among women worldwide and was second in Indonesia. Globally, CC has become the ninth cause of death and the third deadliest cancer in Indonesia [2]. CC affected 9.25 per 100,000 women in Jakarta by 2012 [3]. CC is mainly caused by the human papillomavirus (HPV), and high-risk HPV 16 and 18 (which infections were prevented through vaccinations) contribute to approximately 70% of cancers [4] with devastating effects [2]. Regarding the distribution of cases, a study in the US observed a higher prevalence of CC in urban compared to rural counties (84.5% vs. 15.5%) [5], and in China, a higher HPV prevalence among women in urban (16.3%) than rural areas (13%) [6]. Contradictorily, other studies showed higher prevalence in rural (11%) than urban citizens (10%) due to disparity in vaccination and screening rate among rural residents [7]. Today, no study revealed that comparison in Indonesia, only published data in 2019 showed that the prevalence of HPV infection among urban women in Indonesia was 5.2% [8]. Urban and rural populations live in different socioeconomic backgrounds and lifestyles, explaining the higher prevalence of overall HPV in urban populations. However, there is no existing data on the differences in epidemiology and KAP related to HPV and CC between urban and rural areas of Indonesia. HPV does not only affect women but also men. The HPV prevalence in men is estimated at 1.3%–72.9% globally [9]. According to a report in 2020, the annual crude rate of cancer of the oropharynx, colorectal, lung, liver, nasopharynx, bladder, stomach, esophagus, larynx, lymphatic and blood, brain, oral cavity, and kidney due to HPV infection in Indonesian males was higher than females [10]. Other HPV-related cancers exclusively among male patients were cancers of the prostate (9.85/100,000), testis (1.09/100,000), and penis (0.74/100,000). Moreover, in CC, men played an essential role in helping their wife or daughter participate in CC care and prevention programs. Indeed, the HPV vaccine has been approved for administration to reduce CC and other HPV-related cancers incidence for both men and women [11–13].

Generally, the most prominent factor contributing to the increase of CC and HPV-related cancers across the countries is a poor understanding of causes, risks, and prevention programs for HPV infection [14,15]. Thus, understanding HPV infection, CC, and HPV vaccine are

essential for both sex, notably in the urban community due to their complexity regarding a higher number of violence and crimes, smokers, drugs, economic and social status disparities, and sex workers [16,17]. Approximately 56.7% of Indonesia's population lived in urban areas in 2020 [18]. A greater prevalence of alcoholism, smoking, drug consumption, and risky sexual behavior are risk factors for acquiring HPV infection [19–21].

Through improving knowledge, attitudes, and practices (KAP) among urban communities, we could tackle CC as a significant public health threat among women. Several previous studies have reported that understanding the risk of CC, HPV infection, and HPV vaccine is a strong predictor of vaccination practice or intention to vaccinate [22–24]. However, this topic is scarcely studied in the general population and diverse across various countries [25]. Few studies explored this topic in Indonesia in-depth: a prior study more focused on parent's attitudes, beliefs, and uptake of the HPV vaccination [26], a study only took samples from women in single primary care about knowledge and behavior regarding CC [27], a study was conducted only in junior high school students with tiny sample [28], a study did research on women population about HPV vaccination and CC screening [29], and two studies only assessed knowledge, attitude, and acceptability of HPV vaccination among university students [30,31]. To the best of our knowledge, no reported data on the KAP related to HPV, CC, and vaccination among urban citizens in Indonesia, especially those reporting KAP as a united link of public health construct. Prior studies also did not analyze contributing factors for proper KAP, and did not wholly assessed all three issues of HPV, CC, and vaccination. Therefore, this study attempts to generate information about the association between KAP towards the danger of HPV infection, CC, and HPV vaccination among Indonesian urban citizens, along with contributing factors for appropriate KAP and the readiness of both men and women to receive these vaccinations. This study can assist the government and the medical profession in establishing health policies and taking appropriate measures to increase public' awareness of the risks associated with HPV infection and increase the vaccination rate, which may help effectively prevent HPV infection and CC.

## Materials and methods

### Study design, population, and inclusion criteria

The present study included analyses of data obtained from March 2020 to August 2021 from a cross-sectional survey through an online self-reported questionnaire. The eligibility criteria for participants were urban citizens from diverse backgrounds aged ≥9 years, male or female who had resided in Jakarta, Indonesia, for at least six months [32], had a basic level of literacy and had given written informed consent to participate in the study. Participants who did not completely fill out the questionnaire were excluded. The minimum sample was 196, calculated using a formula of different two proportion calculations in observational studies for descriptive and categorical data with an assumption of 5% alpha (Zα = 1.96), proportion or prevalence of the point of interest category/ condition (P) 0.5 the value of Q resulted from 1-P was 0.5, and absolute level of accuracy or the precision of the estimate was 5% [33,34]. Nevertheless, 400 respondents participated in this study.

### Sampling technique and data collection

Participants were recruited using an exponential non-discriminative Snowball sampling technique. Questionnaires were made in Google Forms and distributed using virtual social media platforms such as Whatsapp, Line, Instagram, and people's networks. The questionnaire was filled out voluntarily by consent and anonymously. After filling the questionnaire, the subjects were approached through text message or called to ensure they were eligible for our inclusion

criteria, mainly to convince them they were Jakarta citizens. The methods carried out in this study were approved by the Research Ethics Committee, Faculty of Medicine, Universitas Indonesia, with letter number KET-237/UN2.F1/ETIK/PPM.00.02/2020.

## Assessment tool and measurement process

The questionnaire was constructed by modifying previously validated questionnaires to match the context of the study [29,35], then translating backward and forward into Bahasa Indonesia. The validity of the new version questionnaire was measured via the comments of five experts in gynecology-oncology. Its reliability was calculated through test-retest with the participation of 30 people (α = 0.8). We assessed KAP regarding HPV, CC, and HPV vaccine. Knowledge and attitudes were grouped into two aspects: understanding of HPV infection and CC (aspect 1); and the HPV vaccination (aspect 2). Practices were evaluated without aspects grouping.

The questionnaire encompassed four primary sections. The first section contained socio-demographical information (8 questions), the second part is for knowledge-related questions (21 questions, comprising nine questions in aspect 1 and 12 questions in aspect 2), the third section is for attitudes-related questions (12 questions, consisting of 5 questions in aspect 1 and 7 questions in aspect 2), and the fourth part is regarding practices (7 questions) [36,37]. The scoring system consists of 3 types of questions based on the answer choices: questions with two answer choices (yes/no), three answer choices (yes/no/do not know or true/false/do not know), or four answer choices (yes/maybe/unsure/no or absolutely agree/agree/disagree/absolutely disagree) adjusted for favorable responses in each item. Referring to the previous study [38], in the type of questions with two and three answer choices, negative answers get a score of 0 while confirming answers get a score of 2. To the kind of question with four answer choices, negative answers get a score of 0, confirming answers get a value of 1, and highly confirming answers get a value of 2. Generally, responses in this questionnaire can be dichotomized as answers reflecting good knowledge/ positive attitude/ favorable (with +1 or +2 points) or poor knowledge/ negative attitude/ unfavorable practices (0 points). The point results of the response will be divided by the maximum score possible in each aspect and multiplied by 100%, resulting in a final score for each aspect and overall. The cut-off for good KAP for the questionnaire is a total score of ≥60% [36,37]. The questionnaires and scoring system is available as supplementary material S1 File.

## Study variables

Independent variables included socio-demographic information (age, education, occupations, salary per month, parent's education level, and religion) and several components of knowledge and attitude towards HPV infection, cervical cancer, and HPV vaccination. The dependent variables encompassed KAP levels towards HPV infection, cervical cancer, and HPV vaccination. However, these dependent variables can play as independent variables compared to each other, such as knowledge (independent) vs. attitude (dependent). Possible confounding variables were government policy regarding HPV vaccination, internet access to health information, the height of pandemic condition, healthcare access facility, cultural, myth, beliefs, and religious influence, as well as caregiving patterns, childbearing, and habits in the family regarding health perception which was not further assessed in this research.

## Statistical analysis

Responses from participants were collected using a spreadsheet linked to the online Google Form questionnaire and exported to the Microsoft® Excel® for Microsoft 365 MSO (Version 2109 Build 16.0.14430.20154) 32-bit to manage and visualize the data. Afterward, data were

analyzed using the Statistical Package for Social Sciences (SPSS) version 24.0. Categorical data were displayed as frequency and percentages, while continuous data were presented as means or medians according to the Kolmogorov-Smirnov test for normality and distribution data measurement. Each item of answers regarding the questionnaire, questionnaire score, and category of quality (poor-good) was analyzed by sex using an appropriate categorical statistic test. We compared men and women to avoid bias for KAP level, and statistical analysis between both sexes revealed no difference in socioeconomic characteristics. The correlation between KAP and intercorrelations among the aspects was investigated with Spearman's correlation test. The value of rho or correlation degree ($\rho$) was decided according to the standard [39]. Additionally, a univariate analysis of potential factors associated with the dependent variables of KAP was done. In the univariate analysis, any factors related to each outcome at $p \leq 0.20$ were considered for inclusion as fixed effects in the step-wise multiple regression analyses model [40]. The final model was developed by performing the backward LR method in the variables selection process [41]. A p-value of $<0.05$ indicated statistical significance. All methods and study results were reviewed and reported following the STROBE reporting guidelines (S1 Checklist) for cross-sectional studies [42].

## Results

We display the socio-demographic characteristics of the study respondents in Table 1. Four hundred respondents (median age: 23 (13–73) years) were recruited from Jakarta Province, one of 34 provinces in Indonesia and the largest metropolitan city in Indonesia. The majority of our respondents of this study were women with high-level education, a bachelor's degree, unemployed, with students most commonly recruited. Most respondents also had wages $\leq$350 USD per month and were Muslim. The highest proportion of education among the respondents' fathers was bachelor's degree with excellent education level, and among respondents' mothers was senior high school (SHS) with low education level. Most participants were young (under 25 years). Their education mainly was bachelor's degree with high educational achievements.

We provide several analyses of questionnaires per item based on sex in supplementary files. S1 Table reports the items of question and answer of knowledge towards HPV infection and CC (aspect 1) and HPV vaccination (aspect 2) with different responses statistically between men and women ($p<0.01$) in 8/9 items in aspect 1 and ($p<0.05$) in 8/12 items in aspect 2. Only 11/21 items were answered by the majority of the participants ($>60\%$) with appropriate responses. S2 Table summarizes participants' attitudes from items related to HPV infection and CC (aspect 1) and HPV vaccination (aspect 2). It showed the different responses statistically between the sexes ($p<0.05$) in 1/5 items in aspect 1 and ($p<0.001$) in 1/7 items in aspect 2. Citizens in this study majorly had positive attitudes, with 10/12 items answered by $>60\%$ participants with appropriate responses. In analyzing the practice question towards HPV infection, CC, and HPV vaccination in S3 Table, 5/7 items of fair practice answered by $>60\%$ participants, and 5/7 items showed different responses statistically between the sexes.

The scoring results in each aspect of KAP are shown in Table 2. The knowledge score in aspect 1 and aspect 2 was still poor (score of $>60$); meanwhile, overall knowledge was considered fair ($>60$); the proportion of poor knowledge is 53.5%, 52.0%, and 49.3%, respectively for both sexes, males, and females. Women's knowledge scores and proportions in all aspects and overall were higher than men's ($p<0.01$). Per aspect or overall, the population had a great attitude (score of $>60$) towards HPV, CC, and the HPV vaccine. The proportion of positive attitudes was 87.3% (aspect 1), 75.0% (aspect 2), and 82.0% (overall attitude aspects). The median score for attitudes was different in women than men ($p<0.05$), except in aspect 1. However,

**Table 1. Socio-demographical characteristics of 400 respondents in Jakarta, Indonesia 2020.**

| Socio-demographic Characteristics | Men (n = 105) | | Women (n = 295) | | Total (n = 400) | |
|---|---|---|---|---|---|---|
| | n | % | n | % | n | % |
| **Age Groups (y.o)** | | | | | | |
| 11–20 | 31 | 29.5 | 92 | 31.2 | 123 | 30.8 |
| 21–30 | 54 | 51.4 | 140 | 47.5 | 194 | 48.5 |
| 31–40 | 6 | 5.7 | 22 | 7.5 | 28 | 7.0 |
| 41–50 | 6 | 5.7 | 23 | 7.8 | 29 | 7.3 |
| 51–60 | 6 | 5.7 | 16 | 5.4 | 22 | 5.5 |
| 61–70 | 2 | 1.9 | 1 | 0.3 | 3 | 0.8 |
| 71–80 | 0 | 0.0 | 1 | 0.3 | 1 | 0.3 |
| Median (Min-Max) | 23.00 (16–62) | | 23.00 (13–73) | | 23.00 (13–73) | |
| **Classification of Young/Old Age** | | | | | | |
| <25 years | 57 | 54.3 | 170 | 57.6 | 227 | 56.8 |
| ≥25 years | 48 | 45.7 | 125 | 42.4 | 173 | 43.3 |
| **Formal Education** | | | | | | |
| Junior High School (JHS) | 3 | 2.9 | 11 | 3.7 | 14 | 3.5 |
| Senior High School (SHS) | 45 | 42.9 | 117 | 39.7 | 162 | 40.5 |
| Bachelor Degree | 48 | 45.7 | 159 | 53.9 | 207 | 51.8 |
| Master Degree | 9 | 8.6 | 6 | 2.0 | 15 | 3.8 |
| Doctoral Degree | 0 | 0.0 | 2 | 0.7 | 2 | 0.5 |
| **Level of Education** | | | | | | |
| Low Education (≤SHS) | 48 | 45.7 | 128 | 43.4 | 176 | 44.0 |
| High Education (≥College) | 57 | 54.3 | 167 | 56.6 | 224 | 56.0 |
| **Occupations** | | | | | | |
| Business and administration professionals | 30 | 28.6 | 88 | 29.8 | 118 | 29.5 |
| Civil servants | 3 | 2.9 | 4 | 1.4 | 7 | 1.8 |
| Health professionals | 3 | 2.9 | 21 | 7.1 | 24 | 6.0 |
| Housewife | 0 | 0.0 | 18 | 6.1 | 18 | 4.5 |
| Independent worker | 10 | 9.5 | 12 | 4.1 | 22 | 5.5 |
| Retired or pensionary worker | 1 | 1.0 | 1 | 0.3 | 2 | 0.5 |
| Student | 57 | 54.3 | 147 | 49.8 | 204 | 51.0 |
| Teacher | 1 | 1.0 | 4 | 1.4 | 5 | 1.3 |
| **Employment** | | | | | | |
| Not working | 58 | 55.2 | 166 | 56.3 | 224 | 56.0 |
| Employed | 47 | 44.8 | 129 | 43.7 | 176 | 44.0 |
| **Salary per month (based on August 29, 2021)** | | | | | | |
| ≤350 USD | 61 | 58.1 | 190 | 64.4 | 251 | 62.8 |
| 351–700 USD | 20 | 19.0 | 59 | 20.0 | 79 | 19.8 |
| 701–1750 USD | 11 | 10.5 | 25 | 8.5 | 36 | 9.0 |
| 1751–3500 USD | 11 | 10.5 | 15 | 5.1 | 26 | 6.5 |
| >3500 USD | 2 | 1.9 | 6 | 2.0 | 8 | 2.0 |
| **Father's Formal Education** | | | | | | |
| No Education | 0 | 0.0 | 1 | 0.3 | 1 | 0.3 |
| Elementary School | 5 | 4.8 | 18 | 6.1 | 23 | 5.8 |
| Junior High School (JHS) | 12 | 11.4 | 26 | 8.8 | 38 | 9.5 |
| Senior High School (SHS) | 19 | 18.1 | 98 | 33.2 | 117 | 29.3 |
| Bachelor Degree | 52 | 49.5 | 105 | 35.6 | 157 | 39.3 |
| Master Degree | 11 | 10.5 | 43 | 14.6 | 54 | 13.5 |

(*Continued*)

**Table 1.** (Continued)

| Socio-demographic Characteristics | Men (n = 105) | | Women (n = 295) | | Total (n = 400) | |
|---|---|---|---|---|---|---|
| | n | % | n | % | n | % |
| Doctoral Degree | 6 | 5.7 | 4 | 1.4 | 10 | 2.5 |
| **Level of Father's Education** | | | | | | |
| Low Education (≤SHS) | 36 | 34.3 | 143 | 48.5 | 179 | 44.8 |
| High Education (≥College) | 69 | 65.7 | 152 | 51.5 | 221 | 55.3 |
| **Mother's Formal Education** | | | | | | |
| No Formal Education | 3 | 2.9 | 1 | 0.3 | 4 | 1.0 |
| Elementary School | 6 | 5.7 | 24 | 8.1 | 30 | 7.5 |
| Junior High School (JHS) | 9 | 8.6 | 32 | 10.8 | 41 | 10.3 |
| Senior High School (SHS) | 40 | 38.1 | 109 | 36.9 | 149 | 37.3 |
| Bachelor Degree | 40 | 38.1 | 108 | 38.6 | 148 | 37.0 |
| Master Degree | 5 | 4.8 | 18 | 6.1 | 23 | 5.8 |
| Doctoral Degree | 2 | 1.9 | 3 | 1.0 | 5 | 1.3 |
| **Level of Mother's Education** | | | | | | |
| Low Education (≤SHS) | 58 | 55.2 | 166 | 56.3 | 224 | 56.0 |
| High Education (≥College) | 47 | 44.8 | 129 | 43.7 | 176 | 44.0 |
| **Religion** | | | | | | |
| Muslim | 46 | 43.8 | 168 | 56.9 | 214 | 53.5 |
| Catholic Christians | 18 | 17.1 | 46 | 15.6 | 64 | 16.0 |
| Protestant Christians | 35 | 33.3 | 63 | 21.4 | 98 | 24.5 |
| Hindu | 0 | 0.0 | 1 | 0.3 | 1 | 0.3 |
| Buddhist | 6 | 5.7 | 14 | 4.7 | 20 | 5.0 |
| Confucian | 0 | 0.0 | 3 | 1.0 | 3 | 0.8 |

the proportion with positive attitudes across aspects and overall was higher for women (p<0.05). Practice related to the understanding of HPV, CC, and vaccination was still low (score of <60). Unfavorable practices were still observed in 69.8% of the population and 84.8% of men.

Spearman's correlation found a correlation between KAP due to abnormality data results in the Kolmogorov-Smirnov test (indicating substantially skewed distribution). Fig 1a, 1c, 1d and 1g show a weak positive correlation between aspect 1 of knowledge and overall attitudes (ρ = 0.301), overall knowledge and overall attitudes (ρ = 0.385), aspect 1 of knowledge and practice (ρ = 0.378), and aspect 1 of attitudes towards practice (ρ = 0.357) respectively. Meanwhile, the moderate positive correlation was observed between the aspect 2 of knowledge and overall attitudes (ρ = 0.409), aspect 2 of knowledge and practice (ρ = 0.515), overall knowledge and practice (ρ = 0.485), aspect 2 of attitudes and practice (ρ = 0.550), and overall attitudes towards practice (ρ = 0.577) as shown in Fig 1b, 1e, 1f, 1h and 1i respectively. All correlation tests had a significant p-value <0.001.

The multivariate logistic regression analysis of several factors contributing to good knowledge is constructed in Table 3. Female sex (OR = 2.99, p<0.001), high-level of respondent's education (OR = 2.91, p<0.01), and a greater mother's educational status (OR = 2.15, p<0.01) are factors of having overall good knowledge towards HPV, CC, and vaccination. Female sex (OR = 3.61, p<0.001), high level of respondent's education (OR = 2.26, p<0.01), mother's educational status (OR = 2.27, p<0.01), and greater wages per month (OR = 2.60, p<0.01) impacted good understanding that CC can be prevented by vaccination. The odds of being aware of HPV vaccine availability was determined by female sex (OR = 3.30, p<0.001), school

**Table 2. Level of knowledge, attitudes, and practice aspects towards HPV, CC, and corresponding vaccine in an urban community Indonesia graded by sex.**

| Parameters | Score based on sex | | | | Total score (n = 400) | | p-value |
|---|---|---|---|---|---|---|---|
| | Men (n = 105) | | Women (n = 295) | | | | |
| | n | % | n | % | N | % | |
| **Knowledge** | | | | | | | |
| **Aspect 1: HPV infection and CC** | | | | | | | |
| Median (Min-Max) | 44.44 (0–100) | | 66.67 (0–100) | | **55.56 (0–100)** | | **0.000**[*a] |
| Poor | 71 | 67.6 | 142 | 48.1 | 213 | **53.3** | **0.001**[#b] |
| Good | 34 | 32.4 | 153 | 51.9 | 187 | 46.8 | |
| **Aspect 2: HPV vaccination** | | | | | | | |
| Median (Min-Max) | 41.67 (8.33–100) | | 66.67 (8.33–100) | | **58.33 (8.33–100)** | | **0.000**[*a] |
| Poor | 71 | 67.6 | 137 | 46.4 | 208 | **52.0** | **0.000**[#b] |
| Good | 34 | 32.4 | 158 | 53.6 | 192 | 48.0 | |
| **Overall Knowledge** | | | | | | | |
| Median (Min-Max) | 42.86 (4.76–100) | | 61.90 (9.52–100) | | **61.90 (4.76–100)** | | **0.000**[*a] |
| Poor | 71 | 67.6 | 126 | 42.7 | 197 | **49.3** | **0.000**[#b] |
| Good | 34 | 32.4 | 169 | 57.3 | 203 | 50.8 | |
| **Attitudes** | | | | | | | |
| **Aspect 1: HPV infection and CC** | | | | | | | |
| Median (Min-Max) | 70.00 (30–100) | | 80.00 (30–100) | | **80.00 (30–100)** | | 0.051[*a] |
| Negative | 22 | 21.0 | 29 | 9.8 | 51 | 12.8 | **0.003**[#b] |
| Positive | 83 | 79.0 | 266 | 90.2 | 349 | **87.3** | |
| **Aspect 2: HPV vaccination** | | | | | | | |
| Median (Min-Max) | 71.43 (21.43–92.86) | | 71.43 (28.57–100) | | **71.43 (21.43–100)** | | **0.023**[*a] |
| Negative | 34 | 32.4 | 66 | 22.4 | 100 | 25.0 | **0.042**[#a] |
| Positive | 71 | 67.6 | 229 | 77.6 | 300 | **75.0** | |
| **Overall Attitudes** | | | | | | | |
| Median (Min-Max) | 70.83 (29.17–91.67) | | 75.00 (33.33–100) | | **70.83 (29.17–100)** | | **0.012**[*a] |
| Negative | 26 | 24.8 | 46 | 15.6 | 72 | 18.0 | **0.036**[#a] |
| Positive | 79 | 75.2 | 249 | 84.4 | 328 | **82.0** | |
| **Practice** | | | | | | | |
| Median (Min-Max) | 42.86 (7.14–92.86) | | 50.00 (7.14–100) | | **50.00 (7.14–100)** | | **0.000**[*a] |
| Unfavorable | 89 | **84.8** | 190 | 64.4 | 279 | **69.8** | **0.000**[#a] |
| Favorable | 16 | 15.2 | 105 | 35.6 | 121 | 30.3 | |

[*]Numerical comparative test;

[#]Categorical comparative test;

[a]Mann-Whitney;

[b]Chi-square;

Percentage of the total column; significant if p<0.05.

level of respondents (OR = 1.86, p<0.05), educational background of fathers (OR = 2.10, p<0.01), and age ≥25 years (OR = 2.54, p<0.01).

Several factors predicting positive attitudes toward HPV, CC, and vaccination were summarized in Table 4. They were respondents' mother's educational background (OR = 4.13, p<0.001), younger age <25 years old (OR = 1.86, p<0.05), and prior good knowledge (OR = 2.96, p<0.001). People who worry about their partner or close family having CC were influenced by their father's educational degree (OR = 2.64, p<0.05) and younger age

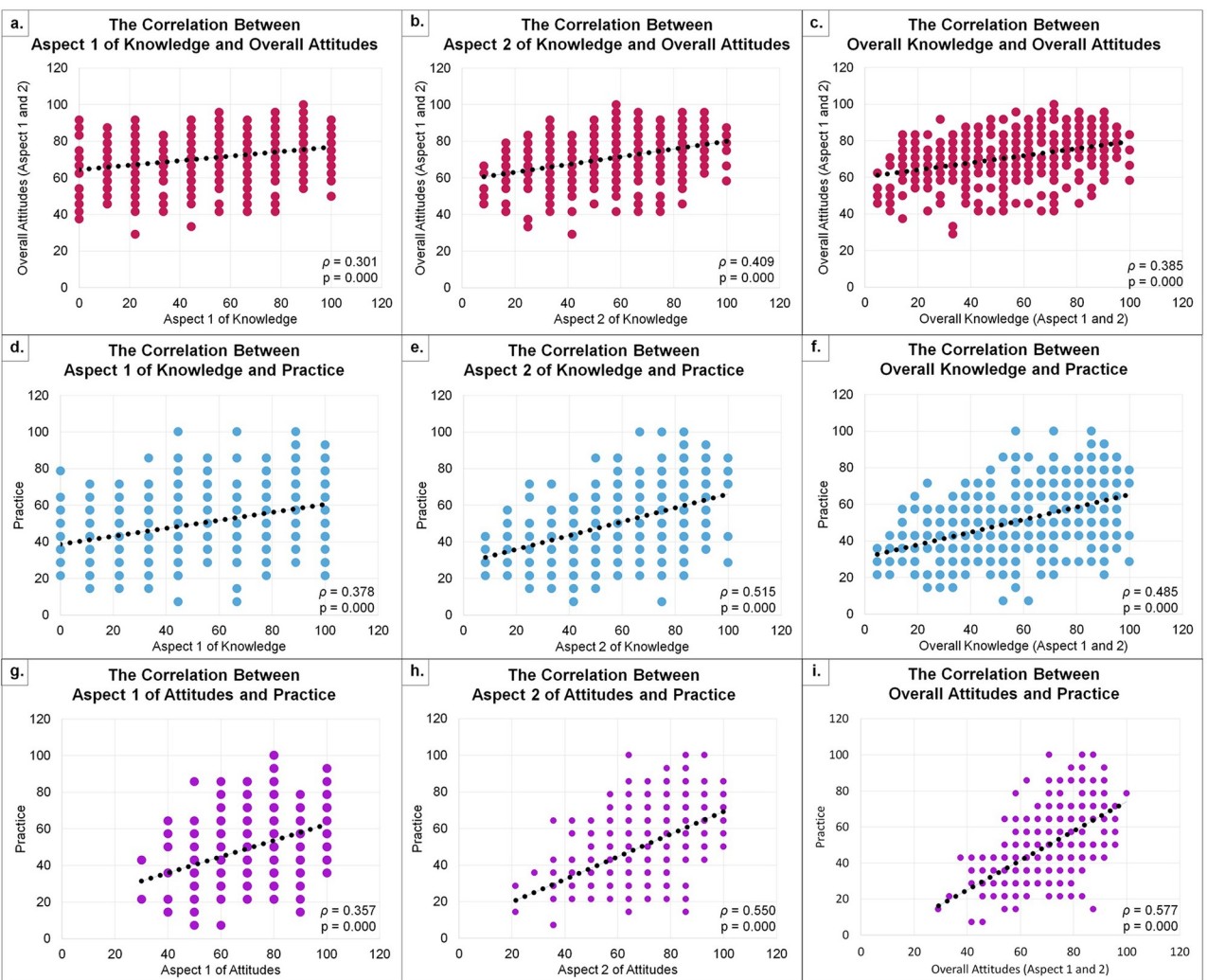

**Fig 1. Correlation between knowledge, attitude, and practice towards HPV, CC, and HPV vaccination.** Aspect 1 is about understanding the HPV virus and CC, while aspect 2 is related to HPV vaccination. Rho-value ($\rho$) 0.2–0.4 is considered as weak positive correlation (charts a, c, d, and g); and 0.4–0.6 is considered as moderate positive correlation (charts b, e, f, h, and i). All statistical analysis using Spearman's test. Normality test with Kolmogorov-Smirnov for aspects 1 and 2 of knowledge, overall knowledge, aspect 1 and 2 of attitude, overall attitude, and overall practice score were 0.000. The data is afterward assumed to be abnormal in the distribution in distribution (indicating substantially skewed distribution).

(OR = 2.52, p<0.05). Younger people were more likely to think that their partner or close family should be vaccinated (OR = 3.59, p<0.001). Respondent's sex, educational level, employment status, and monthly wages did not affect attitudes significantly.

Numerous factors predicting appropriate HPV, CC, and vaccination practices were analyzed in Table 5. Good practice was more likely in the female sex (OR = 2.33, p<0.01), employees (OR = 1.68, p<0.05), people with good knowledge (OR = 4.56, p<0.001), and assertiveness (OR = 8.05, p<0.001). Factors contributing to individuals having at least one dose of the HPV vaccine were being female (OR = 3.88, p<0.05), having a high level of mother's education (OR = 2.42, p<0.01), working with high wage (OR = 2.78, p<0.01), and good knowledge (OR = 2.81, p<0.01). People were more willing to get vaccinated if they were female (OR = 3.48, p<0.001), had excellent knowledge (OR = 2.77, p<0.01), and were a positive attitude in which convinced about the HPV vaccine (OR = 8.04, p<0.001). Respondent's

**Table 3. Multivariate logistic regression analysis for factors towards good knowledge and other related aspects of knowledge regarding HPV, CC, and the vaccine (N = 400).**

| Factors | Overall knowledge towards HPV, CC, and corresponding vaccine | | | | Understanding that CC can be prevented by vaccination | | | | Be aware that HPV vaccine availability in Indonesia | | | |
|---|---|---|---|---|---|---|---|---|---|---|---|---|
| | Unadjusted OR (95%CI) | p-value[a] | Adjusted OR (95%CI) | p-value[b] | Unadjusted OR (95%CI) | p-value[a] | Adjusted OR (95%CI) | p-value[b] | Unadjusted OR (95%CI) | p-value[a] | Adjusted OR (95%CI) | p-value[b] |
| **Sex** | | | | | | | | | | | | |
| Male | Ref | **0.000** | Ref | **0.000** | Ref | **0.000** | Ref | **0.000** | Ref | **0.000** | Ref | **0.000** |
| Female | 2.80 (1.75–4.48) | | 2.99 (1.83–4.87) | | 2.98 (1.86–4.80) | | 3.61 (2.16–6.05) | | 2.63 (1.66–4.17) | | 3.30 (1.99–5.46) | |
| **Respondent's Education** | | | | | | | | | | | | |
| Low Education | Ref | **0.000** | Ref | **0.001** | Ref | **0.000** | Ref | **0.003** | Ref | **0.000** | Ref | **0.020** |
| High Education | 2.51 (1.68–3.77) | | 2.91 (1.89–4.49) | | 2.88 (1.83–4.53) | | 2.26 (1.32–3.86) | | 2.70 (1.75–4.16) | | 1.86 (1.10–3.13) | |
| **Father's Education** | | | | | | | | | | | | |
| Low Education | Ref | **0.018** | Ref | 0.126 | Ref | **0.127** | Ref | 0.263 | Ref | **0.177** | Ref | **0.003** |
| High Education | 1.62 (1.09–2.41) | | 1.52 (0.89–2.61) | | 1.41 (0.91–2.19) | | 1.42 (0.77–2.61) | | 1.34 (0.88–2.04) | | 2.10 (1.30–3.39) | |
| **Mother's Education** | | | | | | | | | | | | |
| Low Education | Ref | **0.006** | Ref | **0.001** | Ref | **0.058** | Ref | **0.002** | Ref | 0.215 | - | n/a |
| High Education | 1.75 (1.17–2.61) | | 2.15 (1.39–3.31) | | 1.55 (0.98–2.43) | | 2.27 (1.37–3.76) | | 1.31 (0.86–2.01) | | - | |
| **Age** | | | | | | | | | | | | |
| <25 years | Ref | 0.294 | - | n/a | Ref | **0.001** | Ref | 0.295 | Ref | **0.000** | Ref | **0.001** |
| ≥25 years | 1.24 (0.83–1.84) | | - | | 2.18 (1.36–3.48) | | 1.45 (0.72–2.90) | | 2.64 (1.68–4.15) | | 2.54 (1.43–4.52) | |
| **Employment** | | | | | | | | | | | | |
| Not working | Ref | 0.253 | - | n/a | Ref | **0.005** | Ref | 0.596 | Ref | **0.006** | Ref | 0.411 |
| Employed | 1.26 (0.85–1.87) | | - | | 1.92 (1.21–3.04) | | 0.82 (0.39–1.75) | | 1.84 (1.19–2.83) | | 0.75 (0.38–1.48) | |
| **Salary per month** | | | | | | | | | | | | |
| Low (<350 USD) | Ref | 0.484 | - | n/a | Ref | **0.000** | Ref | **0.002** | Ref | **0.001** | Ref | 0.422 |
| High (≥350 USD) | 1.16 (0.77–1.74) | | - | | 2.72 (1.63–4.52) | | 2.60 (1.40–4.83) | | 2.18 (1.37–3.46) | | 1.31 (0.68–2.50) | |

[a]Bivariate analysis using Chi-square and Mantel-Haenszel odds ratio estimate, any associated factors with p≤0.20 were deemed eligible for inclusion in the multivariate analysis model;

[b]Multivariate logistic regression analysis;

95%CI (95% confidence intervals).

education, father's education level, and participant's age did not significantly change the respondents' practice, vaccination status, and intent or readiness to take vaccines.

Table 6 describes the KAP-related items' analysis which influenced respondents to get at least one vaccination dose and readiness to get the vaccine. The significant predictors for people who have been vaccinated at least one dose were: (1) knowing the access to get HPV vaccination, (2) understanding the dose recommended, (3) having the willingness to get the vaccine, and (4) not worrying about side effects, all p-value at <0.05. Meanwhile, people intended to and were readier to take the HPV vaccine if they: (1) knew about the protective

**Table 4. Multivariate logistic regression model for factors related to positive attitudes and other related aspects of attitudes towards HPV, CC, and the vaccine (N = 400).**

| Factors | Overall attitudes towards HPV, CC, and corresponding vaccine | | | | Worrying about a partner or closed family will get CC | | | | Thinking about the partner or closed family must be vaccinated against HPV | | | |
|---|---|---|---|---|---|---|---|---|---|---|---|---|
| | Unadjusted OR (95%CI) | p-value[a] | Adjusted OR (95%CI) | p-value[b] | Unadjusted OR (95%CI) | p-value[a] | Adjusted OR (95%CI) | p-value[b] | Unadjusted OR (95%CI) | p-value[a] | Adjusted OR (95%CI) | p-value[b] |
| **Sex** | | | | | | | | | | | | |
| Male | Ref | **0.036** | Ref | 0.149 | Ref | **0.119** | 1.95 | 0.073 | Ref | 0.819 | - | n/a |
| Female | 1.78 (1.04–3.07) | | 1.53 (0.86–2.74) | | 1.74 (0.86–3.50) | | (0.94–4.05) | | 0.93 (0.48–1.78) | | - | |
| **Respondent's Education** | | | | | | | | | | | | |
| Low Education | Ref | **0.080** | Ref | 0.266 | Ref | **0.021** | Ref | 0.244 | Ref | **0.012** | Ref | 0.939 |
| High Education | 0.62 (0.37–1.06) | | 0.68 (0.34–1.35) | | 0.42 (0.19–0.89) | | 0.59 (0.24–1.44) | | 0.46 (0.25–0.85) | | 0.97 (0.43–2.17) | |
| **Father's Education** | | | | | | | | | | | | |
| Low Education | Ref | **0.005** | Ref | 0.992 | Ref | **0.002** | Ref | **0.011** | Ref | **0.044** | Ref | 0.743 |
| High Education | 2.09 (1.25–3.52) | | 1.03 (0.52–1.95) | | 2.96 (1.45–6.05) | | 2.64 (1.25–5.59) | | 1.79 (1.01–3.16) | | 1.13 (0.55–2.32) | |
| **Mother's Education** | | | | | | | | | | | | |
| Low Education | Ref | **0.000** | Ref | **0.000** | Ref | **0.003** | Ref | 0.458 | Ref | **0.005** | Ref | 0.154 |
| High Education | 4.04 (2.17–7.53) | | 4.13 (2.21–7.73) | | 3.25 (1.45–7.23) | | 1.48 (0.53–4.18) | | 2.41 (1.28–4.51) | | 1.63 (0.83–3.19) | |
| **Age** | | | | | | | | | | | | |
| ≥25 years | Ref | **0.002** | Ref | **0.031** | Ref | **0.001** | Ref | **0.014** | Ref | **0.000** | Ref | **0.000** |
| <25 years | 2.26 (1.34–3.80) | | 1.86 (1.06–3.28) | | 3.17 (0.16–6.48) | | 2.52 (1.2–5.28) | | 3.59 (1.96–6.61) | | 3.59 (1.96–6.61) | |
| **Employment** | | | | | | | | | | | | |
| Not working | Ref | **0.055** | Ref | 0.512 | Ref | **0.004** | Ref | 0.774 | Ref | **0.003** | Ref | 0.775 |
| Employed | 0.61 (0.36–1.01) | | 1.28 (0.61–2.68) | | 0.37 (0.18–0.75) | | 0.87 (0.33–2.28) | | 0.42 (0.23–0.75) | | 1.12 (0.51–2.50) | |
| **Salary per month** | | | | | | | | | | | | |
| Low (<350 USD) | Ref | **0.053** | Ref | 0.806 | Ref | **0.006** | Ref | 0.529 | Ref | **0.000** | Ref | 0.113 |
| High (≥350 USD) | 0.60 (0.36–1.01) | | 1.10 (0.53–2.26) | | 0.39 (0.19–0.782) | | 0.75 (0.31–1.82) | | 0.33 (0.18–0.58) | | 0.56 (0.28–1.15) | |
| **Overall Knowledge** | | | | | | | | | | | | |
| Poor | Ref | **0.000** | Ref | **0.000** | Ref | **0.144** | Ref | 0.290 | Ref | 0.682 | - | n/a |
| Good | 3.03 (1.74–5.26) | | 2.96 (1.67–5.27) | | 1.66 (0.84–3.28) | | 1.47 (0.72–3.00) | | 1.13 (0.64–1.98) | | - | |

[a]Bivariate analysis using Chi-square and Mantel-Haenszel odds ratio estimate, any associated factors with p≤0.20 were deemed eligible for inclusion in the multivariate analysis model;

[b]Multivariate logistic regression analysis;

95%CI (95% confidence intervals).

effect of the HPV vaccine, (2) perceived the dangers of HPV, (3) wanted to get vaccinated, (4) had no fear of HPV vaccination, (4) wanted to share their knowledge and willingness to their surroundings, and (5) intended to get information more about HPV, CC, and the HPV vaccine, all p-value at <0.05. There were 17 items of knowledge, 6 points of attitudes, and three

**Table 5. Multivariate logistic regression model for factors favorable/appropriate practices and others related aspect of practices towards HPV, CC, and corresponding vaccine (N = 400).**

| Factors | Favorable practice towards HPV, CC, and corresponding vaccine | | | | Have been vaccinated against HPV at least one dose | | | | Intention and readiness to take HPV vaccination | | | |
|---|---|---|---|---|---|---|---|---|---|---|---|---|
| | Unadjusted OR (95%CI) | p-value[a] | Adjusted OR (95%CI) | p-value[b] | Unadjusted OR (95%CI) | p-value[a] | Adjusted OR (95%CI) | p-value[b] | Unadjusted OR (95%CI) | p-value[a] | Adjusted OR (95%CI) | p-value[b] |
| **Sex** | | | | | | | | | | | | |
| Male | Ref | **0.000** | Ref | **0.009** | Ref | **0.000** | Ref | **0.003** | Ref | **0.000** | Ref | **0.000** |
| Female | 3.07 (1.72–5.51) | | 2.33 (1.23–4.41) | | 4.48 (1.88–10.69) | | 3.88 (1.57–9.57) | | 4.18 (2.37–7.39) | | 3.48 (1.82–6.65) | |
| **Respondent's Education** | | | | | | | | | | | | |
| Low Education | Ref | **0.007** | Ref | 0.515 | Ref | 0.153 | Ref | 0.430 | Ref | 0.463 | - | n/a |
| High Education | 1.83 (1.17–2.85) | | 1.23 (0.66–2.27) | | 1.48 (0.86–2.53) | | 0.74 (0.35–1.57) | | 1.23 (0.71–2.13) | | - | |
| **Father's Education** | | | | | | | | | | | | |
| Low Education | Ref | 0.118 | Ref | 0.901 | Ref | 0.118 | Ref | 0.698 | Ref | 0.147 | Ref | 0.307 |
| High Education | 1.41 (0.92–2.18) | | 1.04 (0.55–1.97) | | 1.53 (0.87–2.62) | | 0.86 (0.40–1.85) | | 1.50 (0.87–2.60) | | 1.41 (0.73–2.74) | |
| **Mother's Education** | | | | | | | | | | | | |
| Low Education | Ref | **0.018** | Ref | 0.248 | Ref | **0.002** | Ref | **0.004** | Ref | **0.037** | Ref | 0.928 |
| High Education | 1.67 (1.09–2.57) | | 1.35 (0.81–2.24) | | 2.29 (1.35–3.89) | | 2.42 (1.32–4.42) | | 1.85 (1.03–3.32) | | 0.96 (0.41–2.25) | |
| **Age** | | | | | | | | | | | | |
| <25 years | Ref | 0.213 | - | n/a | Ref | 0.267 | - | n/a | Ref | 0.389 | - | n/a |
| ≥25 years | 1.31 (0.86–2.02) | | - | | 1.34 (0.80–2.26) | | - | | 0.79 (0.45–1.36) | | - | |
| **Employment** | | | | | | | | | | | | |
| Not working | Ref | 0.089 | Ref | **0.036** | Ref | 0.133 | Ref | 0.615 | Ref | 0.910 | - | n/a |
| Employed | 1.45 (0.94–2.23) | | 1.68 (1.04–2.72) | | 1.49 (0.89–2.50) | | 1.20 (0.59–2.47) | | 1.03 (0.59–1.80) | | - | |
| **Salary per month** | | | | | | | | | | | | |
| Low (<350 USD) | Ref | 0.182 | Ref | 0.351 | Ref | **0.023** | Ref | **0.001** | Ref | 0.633 | - | n/a |
| High (≥350 USD) | 1.35 (0.87–2.08) | | 1.36 (0.72–2.54) | | 1.83 (1.08–3.08) | | 2.78 (1.53–5.06) | | 0.87 (0.50–1.53) | | - | |
| **Overall Knowledge** | | | | | | | | | | | | |
| Poor | Ref | **0.000** | Ref | **0.000** | Ref | **0.000** | Ref | **0.001** | Ref | **0.000** | Ref | **0.005** |
| Good | 5.79 (3.52–9.50) | | 4.56 (2.72–7.63) | | 3.99 (2.19–7.28) | | 2.81 (1.50–5.28) | | 4.58 (2.39–8.78) | | 2.77 (1.37–5.63) | |
| **Overall Attitudes** | | | | | | | | | | | | |
| Negative | Ref | **0.000** | Ref | **0.000** | Ref | **0.004** | Ref | 0.075 | Ref | **0.000** | Ref | **0.000** |
| Positive | 9.43 (3.35–26.49) | | 8.05 (2.75–23.56) | | 4.20 (1.48–11.94) | | 2.72 (0.90–8.17) | | 9.43 (5.14–17.33) | | 8.04 (4.18–15.45) | |

[a]Bivariate analysis using Chi-square and Mantel-Haenszel odds ratio estimate, any associated factors with p≤0.20 were deemed eligible for inclusion in the multivariate analysis model;

[b]Multivariate logistic regression analysis;

95%CI (95% confidence intervals).

**Table 6. Multivariate logistic regression model for the related aspect of KAP for getting vaccinated minimal one dose and willingness to get vaccinated among respondents (N = 400).**

| The aspect of Knowledge, Attitudes, and Practice | Have been vaccinated against HPV at least one dose | | | | Intention and readiness to take HPV vaccination | | | |
|---|---|---|---|---|---|---|---|---|
| | Unadjusted OR (95%CI) | p-value[a] | Adjusted OR (95%CI) | p-value[c] | Unadjusted OR (95%CI) | p-value[a] | Adjusted OR (95%CI) | p-value[c] |
| **Good knowledge of:** | | | | | | | | |
| Etiology of CC | 3.29 (1.37–7.89) | 0.005 | 0.72 (0.19–2.79) | 0.639 | 1.75 (0.95–3.23) | 0.072 | 0.70 (0.27–1.79) | 0.456 |
| Overview and prior knowledge of HPV | 3.68 (1.86–7.26) | 0.000 | 1.21 (0.45–3.28) | 0.704 | 2.85 (1.63–4.99) | 0.000 | 1.60 (0.71–3.60) | 0.254 |
| HPV related malignancy | 2.93 (1.72–4.99) | 0.000 | 1.50 (0.74–3.02) | 0.257 | 1.68 (0.87–3.24) | 0.116 | 0.81 (0.28–2.37) | 0.694 |
| Transmission of HPV in men | 2.35 (1.34–4.12) | 0.002 | 0.73 (0.29–1.85) | 0.504 | 2.56 (1.44–4.54) | 0.001 | 1.18 (0.37–3.74) | 0.776 |
| Risk of HPV infection in men | 1.62 (0.97–2.73) | 0.067 | 0.59 (0.31–1.13) | 0.133 | 1.70 (0,95–3.06) | 0.072 | 0.71 (0.28–1.80) | 0.466 |
| Sex as a way to transmit HPV | 2.86 (1.41–5.80) | 0.003 | 1.71 (0.65–4.46) | 1.706 | 2.06 (1.17–3.62) | 0.011 | 1.49 (0.62–3.56) | 0.375 |
| The abundance of HPV infection | 3.22 (1.80–5.74) | 0.000 | 0.63 (0.27–1.48) | 0.289 | 1.73 (0.99–3.02) | 0.053 | 0.50 (0.23–1.12) | 0.093 |
| HPV can causes CC | 6.45 (2.28–18.19) | 0.000 | 1.49 (0.36–6.28) | 0.585 | 2.61 (1.47–4.64) | 0.001 | 1.32 (0.40–4.41) | 0.649 |
| Smoking as a risk of HPV infection | 1.86 (1.08–3.18) | 0.023 | 1.39 (0.72–2.70) | 0.325 | 1.79 (0.91–3.50) | 0.088 | 1.30 (0.49–3.42) | 0.601 |
| Vaccine to prevent CC | 16.22 (3.90–67.46) | 0.000 | 3.18 (0.68–14.97) | 0.143 | 1.81 (1.02–3.21) | 0.042 | 0.53 (0.22–1.28) | 0.155 |
| Availability of HPV vaccines | 6.21 (2.61–14.77) | 0.000 | 1.25 (0.36–4.29) | 0.724 | 2.25 (1.29–3.92) | 0.004 | 1.32 (0.48–3.64) | 0.592 |
| **Accessibility to get HPV vaccines** | 7.51 (3.16–17.84) | 0.000 | **3.20 (1.26–8.13)** | **0.014** | 2.33 (1.34–4.05) | 0.002 | 1.21 (0.49–3.00) | 0.681 |
| Side effect of HPV vaccines | 0.97 (0.51–1.84) | 0.917 | - | n/a | 1.20 (0.59–2.42) | 0.617 | - | n/a |
| Risk of infection after vaccination | 2.20 (1.29–3.77) | 0.003 | 0.82 (0.38–1.74) | 0.600 | 2.92 (1.58–5.37) | 0.000 | 1.86 (0.84–4.13) | 0.125 |
| Neccesity of HPV vaccine for infected-people | 1.02 (0.59–1.75) | 0.957 | - | n/a | 1.54 (0.83–2.84) | 0.166 | 1.02 (0.42–2.48) | 0.973 |
| Screening still needed for HPV-vaccinated people | 2.61 (1.24–5.46) | 0.009 | 1.18 (0.46–3.03) | 0.731 | 2.89 (1.64–5.11) | 0.000 | 1.41 (0.59–3.39) | 0.437 |
| **The protection of HPV vaccination** | 2.78 (1.58–4.89) | 0.000 | 0.95 (0.43–2.08) | 0.900 | 2.36 (1.32–4.20) | 0.003 | **2.71 (1.25–5.89)** | **0.012** |
| Age-recommended for HPV vaccination | 6.39 (3.07–13.30) | 0.000 | 1.75 (0.73–4.19) | 0.212 | 2.47 (1.41–4.35) | 0.001 | 1.39 (0.57–3.40) | 0.464 |
| **Dose-recommended for HPV vaccination** | 6.44 (3.26–12.71) | 0.000 | **3.15 (1.50–6.58)** | **0.002** | 3.20 (1.76–5.84) | 0.000 | 1.12 (0.41–3.04) | 0.826 |
| Not to having multiple sex partners after vaccination | 3.14 (1.55–6.37) | 0.001 | 0.81 (0.28–2.29) | 0.686 | 3.04 (1.73–5.32) | 0.000 | 1.97 (0.83–4.71) | 0.127 |
| **Positive attitude of:** | | | | | | | | |
| **HPV infection perceived as a dangerous virus** | 6.68 (1.59–28.06) | 0.003 | 3.55 (0.77–16.36) | 0.104 | 3.31 (1.74–6.31) | 0.000 | **2.89 (1.22–6.87)** | **0.016** |
| HPV infection susceptibility | 1.23 (0.71–2.13) | 0.455 | - | n/a | 2.52 (1.23–5.16) | 0.009 | 1.51 (0.62–3.69) | 0.362 |
| Worries about CC risk in partner and family | 1.42 (0.53–3.77) | 0.483 | - | n/a | 4.00 (1.94–8.30) | 0.000 | 2.73 (0.98–7.61) | 0.055 |
| Support vaccination for children | 0.72 (1.47–3.56) | 0.657[b] | - | n/a | 2.93 (0.71–12.05) | 0.139[b] | 0.91 (0.10–8.64) | 0.934 |
| Support vaccination for adults | indefinite | 0.223[b] | - | n/a | 4.42 (0.96–20.28) | 0.072[b] | 1.43 (0.13–15.98) | 0.769 |
| **Willingness to get HPV vaccines** | 7.13 (2.18–23.29) | 0.000 | **4.73 (1.36–16.50)** | **0.015** | 9.13 (5.01–16.63) | 0.000 | **6.61 (3.16–13.83)** | **0.000** |
| Suggestion of vaccination for partner and family | 0.83 (0.41–1.70) | 0.609 | - | n/a | 4.22 (2.23–7.99) | 0.000 | 1.10 (0.37–3.27) | 0.871 |
| Not fear of injection in general | 1.97 (1.03–3.77) | 0.037 | 0.94 (0.43–2.09) | 0.882 | 1.63 (0.92–2.88) | 0.093 | 0.80 (0.35–1.84) | 0.603 |
| **No worriness about side effect of HPV vaccines** | 7.66 (3.88–15.12) | 0.000 | **4.63 (2.25–9.54)** | **0.000** | 2.07 (1.16–3.69) | 0.012 | 1.82 (0.82–4.04) | 0.140 |
| **No fear of HPV vaccination** | 13.26 (1.80–97.54) | 0.001 | 6.75 (0.86–53.18) | 0.070 | 4.85 (2.56–9.17) | 0.000 | **3.65 (1.59–8.37)** | **0.002** |
| **Favorable practice of:** | | | | | | | | |
| Decision is controled by parents | 1.86 (1.09–3.16) | 0.022 | 0.69 (0.36–1.33) | 0.269 | 0.85 (0.49–1.46) | 0.547 | - | n/a |
| Decision is influenced by friends | 1.09 (0.57–2.12) | 0.791 | - | n/a | 0.77 (0.37–1.60) | 0.484 | - | n/a |
| Religions support HPV vaccination | 1.61 (0.70–3.72) | 0.259 | - | n/a | 2.05 (1.04–4.03) | 0.035 | 1.09 (0.37–3.17)/ | 0.878 |

*(Continued)*

**Table 6.** (Continued)

| The aspect of Knowledge, Attitudes, and Practice | Have been vaccinated against HPV at least one dose | | | | Intention and readiness to take HPV vaccination | | | |
|---|---|---|---|---|---|---|---|---|
| | Unadjusted OR (95%CI) | p-value[a] | Adjusted OR (95%CI) | p-value[c] | Unadjusted OR (95%CI) | p-value[a] | Adjusted OR (95%CI) | p-value[c] |
| Sharing knowledge about HPV and CC | indefinite | 0.012[b] | - | n/a | 20.39 (8.04–51.68) | 0.000 | **12.36 (3.78–40.43)** | **0.000** |
| Intention to know more about HPV, CC, vaccine | 1.70 (0.38–7.58) | 0.750[b] | - | n/a | 18.53 (6.32–54.33) | 0.000[b] | **9.17 (2.29–36.71)** | **0.002** |

[a]Bivariate analysis using Chi-square and [b]Fisher's exact test which any associated factors with p≤0.20 were deemed eligible for inclusion in the multivariate analysis model; [c]Multivariate logistic regression analysis; Indefinite results caused by the presence of zero proportion. CC, Cervical cancer; 95%CI (95% confidence intervals).

types of practice in the model, which did not affect the status of having been vaccinated for one dose or more HPV vaccine and intention or readiness to be injected with the HPV vaccine.

## Discussion

Literacy, screening, and vaccination become a practical approaches in diminishing the burden of CC. However, the KAP of Indonesian people about this issue in previous studies was poor [29,30,43]. Thereby, we aim to reevaluate the KAP towards HPV, CC, and vaccine, along with socio-demographic characteristics influences in an urban community where a massive CC prevention program has been established.

### Socio-demographic characteristic

In this study, the participants are prominently young people (21–30 years) with a proportion of 48.5%, aligning with the targets of the HPV vaccine. Most participants and their fathers graduated from college, while the proportion of respondents' mothers who finished college was lower than their fathers. Reflecting data in urban areas of Indonesia, 2020 [18], the college education level was still low (12.79%) compared to SHS (35.44%) as a dominant education level [44]. Most respondents were students, while the dominant workers were business and administration professionals. This information is similar to data in Jakarta, in which employees in the administration and professional sectors are dominant (2.73 million), followed by independent workers (1.14 million) [45]. The monthly salary of most participants in this study was approximately under 350 USD due to a minimal standard of wages in Jakarta of around 310 USD (4,416,185.55 IDR) per month, while generally, Indonesian had a general minimum wage of 188 USD (2,674,691.82 IDR) per month in 2020–2021 [46]. The monthly payment in our country is relatively low compared to other neighboring countries in Southeast Asia (Singapore, Malaysia, Thailand, and the Philippines) [47]. Financial status was introduced as an influential factor to KAP toward HPV, CC, and vaccination [48]. Participants mostly come from Muslim populations aligning with the characteristics of the Indonesian people. The influence of religion on HPV, CC, and vaccination is still conflicting [49]. In Asia, home to the most populous Muslim community, the principal concerns are non-halal materials in vaccines, particularly porcine or porcine-derived components. Still, considering the advantages and disadvantages, quadrivalent HPV vaccination is routinely suggested in Indonesia, similar to Malaysia, another preceding Muslim country that had implemented it since 2010, as it is safe, halal, and efficacious to prevent four HPV types of infection [50]. Our study also observed no religion's effect in the analysis. A previous study among Indonesian parents perceived that religion did not affect attitudes toward the HPV vaccination [51].

## The correlation between knowledge, attitudes, and practice aspects

Based on Fig 1, the positive correlations between overall knowledge-attitude (weak), overall knowledge-practice (moderate), and overall attitude-practice (moderate) in this study reaffirm that the relationship between KAP towards an understanding of HPV, CC, and HPV vaccination is not negligible. Moreover, we further analyzed the correlation of each aspect to overall KAP. A weak correlation was observed between knowledge of HPV and CC to overall attitudes and practice. Knowledge regarding HPV vaccination moderately correlates to overall attitude and practice. Meanwhile, the practice was weakly correlated towards attitudes regarding HPV infection and CC and moderately correlated to the perception of the HPV vaccine. The findings align with the results presented by Nurjihan et al. in 2019 [52]. Analysis of each aspect highlights that knowledge and attitude about the HPV vaccine (aspect 2) are more relevant than understanding HPV infection and CC (aspect 1) to improving the HPV vaccination program. With a higher level of knowledge, people will perceive positive attitudes, and with more excellent attitudes, respondents will be more motivated to practice prevention related to HPV-related disease and CC.

We found significant differences in sex regarding all aspects of KAP inquired in this study. Our results revealed that women participants had a more excellent score than men supporting the previous survey [53]. The proportion adjusted by sex is also different, with significantly higher proportions of good KAP in female populations. The knowledge about HPV, CC, and HPV vaccination was terrible, especially in men, possibly due to a lack of awareness and low encouragement to seek information about this topic, which is more relatable to women in their perspective. Still, although women in Indonesia showed a better KAP in receiving and asking for the HPV vaccine, they were more likely to be less familiar with it. This is possibly due to several barriers, including beliefs and cultural restrictions [54]. Moreover, in our country, little to no encouragement from partners and parents and a lack of collaborative schemes from health and other community sectors may limit women from having CC prevention programs. This is corroborated by a preceding study [52].

Our findings demonstrated that 70.9% of women still do not know about smoking as one of the risk factors for HPV infection. Almost 90% of women also do not acknowledge about side effects of the HPV vaccine. Subsequently, only one-third of women considered themselves susceptible to HPV infection. Besides, half of the women still worry about the side effects of the HPV vaccine. Moreover, our research shows that almost half of respondents do not recognize that men can transmit HPV, corroborated by studies in Malaysia [50,55]. It is probably due to unawareness, where Indonesia's HPV and CC-related program focused on all females. This issue could be a barrier to HPV vaccine acceptance if it is to be enforced among boys in Indonesia. Interestingly, in Turkey [56], parents are more eager to vaccinate their sons than daughters. However, there was a demand to enhance the literacy of young males about HPV and its associated diseases and the advantages of the vaccination [57].

## Knowledge level and contributing factors to knowledge

Lack of proper knowledge concerning the role of HPV in the causation of CC and the corresponding vaccine is one of the most prominent settling factors and is responsible for the lower vaccination uptake [58]. The knowledge responses regarding HPV, CC, and HPV vaccinations in our research were not satisfying. Approximately 49.3% of people are still poor in understanding HPV, CC, and corresponding vaccines. Looking up to more specific aspects, only 55.3% of participants demonstrated good knowledge regarding HPV infection and CC (aspect 1). Also, only 52.0% of the participants revealed excellent knowledge regarding HPV vaccination (aspect 2). Although the participants recruited in this study are from the urban

community, the knowledge level was unsatisfactory, corroborated by previous studies in the metropolitan area of India [59,60]. However, the results were better than the study conducted in Chinese women [61–63] and poorer than those in Australia [64] and the UK [65]. Most of the participants who have been aware of the vaccine availability in Indonesia are educated through the Internet, magazine, or other mass media similar to prior studies [53,66].

Regular screening is also still needed even though people have been vaccinated. In our study, the understanding of the importance of screening is still relatively low. Currently, World Health Organization (WHO) prefers using HPV DNA detection as the principal screening tool to visual inspection with acetic acid (VIA) or cytology in screening and treatment approaches among women, either with triage or without triage. WHO advises using partial genotyping, unaided VIA, cytology including the Papanicolaou smear test and liquid-based cytology, or colposcopy that may or may not include a biopsy to triage women after a positive HPV DNA test. The CC screening and treating approach starts at the age of 30 and is repeated regularly every 5–10 years when using HPV DNA detection [67]. However, in Indonesia, HPV DNA access is not widely available and is still far behind the latest recommendations.

In the logistic regression model described in Table 3, we found that the sex (female), respondent's education level, and level of mother's education were predictors of knowledge about HPV, CC, and the vaccine similar to the previous study [68]. Furthermore, understanding of the prevention of CC as being enabled by the vaccine was influenced by sex (female), respondent's education level, mother's education level, and salary per month, similar to prior research [68]. The higher and more positive these factors are, the better the outcome on knowledge. Awareness about the availability of vaccines was influenced by sex (female), participant's education (college graduates), father's level of education (completed college), and older age ($\geq$25 years).

A Chinese study also observed an association between education and knowledge towards HPV, CC, and the vaccine [69]. It is observed that the mother's educational level was more influential for her children's knowledge about CC prevention by vaccination, similar to previous studies [70,71]; meanwhile, the father's education background was prominent in awareness of vaccine availability. Fathers in Indonesia are leaders who generally decide what health programs are allowed to participate by their children and become financial supporters to pay for the vaccine. Vaccine availability awareness is a critical issue; if the individual and parents remain uneducated about the access and availability, the intention and adherence to vaccination will stay low. This elucidates that parents' education is imperative as a foundation to transfer knowledge for their children and become a bridge for appropriate attitudes and practices. Employee status did not affect HPV, CC, and vaccination knowledge status. Students who have not worked can still gain optimal knowledge and participate in HPV infection prevention programs. Salary became a factor of vaccination understanding, but not in awareness of vaccine availability.

The excellent predictor for being aware of HPV vaccine availability is age $\geq$25 years. In Indonesia, the older generation is more familiar with HPV vaccines than the youth. In contrast, the CC prevention program by vaccination is effectively open only to school-age and adolescents who have not had sex. Indonesia's low level of awareness should be improved by disseminating print and electronic media as a reference of medical learning for the new generation [25]. The Internet and social media have been widely used to explore health-related information in multiple countries [72,73]. Thus, Indonesian healthcare providers should endeavor to create guidelines to help young adults appraise the content and the quality of medical information from the Internet [25]. Also, it illustrates a need for a strategy for colleges and

   

governmental bodies to promote appropriate platforms (i.e., Web) for disseminating HPV, CC, and HPV vaccine-related knowledge to scholars and the broad population.

## Attitude level and contributing factors to attitudes

Our study highlights the positive attitude of the sample, 82% overall, 87.3% regarding HPV and CC, and 75% relating HPV vaccination. The median score for attitudes was good overall and for each aspect. Many respondents support using the HPV vaccine both in children and adults. This issue is critical since children can not determine their health issues independently and still rely on their parents to allow vaccination. However, a third of people professed fear of injections, and a half of participants were worried about vaccine side effects.

Multivariate logistic regression analysis summarized in Table 4 described several factors predicting positive attitudes towards HPV, CC, and the vaccination, such as level of mother's education, younger age (<25 years), and good knowledge. People who worry about their partner or close family will have CC were influenced by a level of father's education and younger age. Younger adults below 25 years were more likely to think that their partner or close family should be vaccinated. The prior knowledge has been described earlier in the correlation context that the higher knowledge participants have, thus more positive their attitudes regarding HPV, CC, and vaccination should be, as is corroborated in previous research [52].

The mother's education level was a strong determinant of their children's attitude overall. Meanwhile, the respondent's father's educational level was a notable factor to worry that the respondents' partner and close family will get CC. Fathers in the family play an essential part as role models and should participate in the HPV vaccination decision-making process to support daughters to get the vaccine. In Indonesian culture, the husband's opinion was necessary when the mother decided about health-related issues on children, similar to the culture found in Japan and other Asian countries [74]. Presumably, if the fathers are poor-informed or have negative beliefs about the HPV vaccine, the mother's plan to vaccinate their daughters would not be achieved [74].

Young age became a predictor of good attitude, which generally might be influenced by the source of information which makes their attitude more positive. Age was an influential factor in participants' beliefs regarding CC, where young people were more likely to have reasonable beliefs about vaccination than the elderly [24]. Some of the older population in Indonesia adhere to mistaken beliefs that vaccination is harmful and can lead to infertility. Although parents, especially mothers, play a significant role in increasing the knowledge and attitudes about HPV, accepting health messages from family members or obtaining free and invalid ("fake") broadcasted information from social media groups also has a potentially harmful influence on medical practice as is seen in Indonesian today [25]. Thus, fake news may potentially miss the chance to promote vaccination [75]. Zhuang et al. [76] found a notable positive association between obtaining accurate HPV vaccination information from family members and frequency of HPV vaccination, implying beneficial consequences for children as long as the spread information is valid.

## Practice level and contributing factors to practices

Principally, CC and other HPV-related diseases are preventable by vaccine [77]. The prevalence of HPV infection decreased by 56% among females aged 14–19 and 88% among females and males aged 18–33 after the vaccine was introduced [78]. HPV vaccination in Indonesia was commenced in 2012, with Jakarta becoming one of the cities that have implemented HPV vaccination since 2016, followed by Yogyakarta, Surabaya, Manado, and Makassar in 2017 [79]. The program targets 5th-grade female students in primary school and 6th-grade for the

following dose in the immunization month for school children. In other provinces of Indonesia, the HPV vaccine is still included as elective immunization that all teenagers might not receive and must be purchased on their initiative [54,80,81]. Today, both bivalent and quadrivalent vaccines were licensed, but a quadrivalent vaccine became the recommended prophylactic agent implemented in Indonesia with intervals 0 and 12 months for 9–13 years children [80,81]. Indonesia's government still advised catch-up vaccination with three doses of bivalent vaccine for 14–25 years adults or quadrivalent vaccine for 14–45 years people [82]; meanwhile, a nonavalent vaccine was still not introduced.

The age target differs across the countries. In the US, Food and Drug Administration (FDA) and The Advisory Committee on Immunization Practices (ACIP) have advised HPV vaccination for 9–26 years females and males regardless of sexual activity and HPV DNA status [80,83,84]. It should be started before their 15[th] birthday with 6–12 months apart for two doses and three doses (0,1–2, and 6 months interval) for young adults, teenagers, and immunocompromised persons [85]. ACIP did not recommend catch-up vaccination for all ages between 27–45 [83,86]. Similarly, in the UK, children aged 12–13 years (school year 8) are routinely offered a two-dose HPV vaccine schedule at 0 and 6–24 months [87]. People who have the first dose of the HPV vaccine at ≥15 years, immunosuppressed individuals, or those who have HIV positive will need three doses of vaccination [88]. The vaccine service is available for free on the National Health Service until age 25. In India, the government targeted the 9–14 years old age group but is licensed to use for 9–45 years old patients with two-dose for girls <15 years and three doses for women ≥15 years, immunocompromised patients, patients catching up on vaccination, and in older age (≥26 years). India has a similar policy with Indonesia where HPV vaccine is not prioritized for males due to the restricted budget and consideration of CC burden still being higher among females [89,90]. Contrary, Australia has become the first country to offer the quadrivalent HPV vaccine for males aged 9–26 years within the school-based Australian National Immunisation Program for girls and boys aged 12–13 years in a 2-dose schedule [90]. Meanwhile, in Thailand, the HPV vaccination program involves males aged 13–21 years besides females aged 9–26 years. Indeed, catch up vaccination to age 26 is suggested for men who have sex with men (MSM), transgender people, and immunocompromised persons [91]

Despite the established efficacy and potency for enormous positive impact at the community level, the uptake of HPV vaccination is deficient in various countries, including Indonesia [31,43,54]. By 2017, vaccination coverage in Indonesia among the 5th-grade primary school students was 70.9%, still centralized in an urban area [43]. However, these results indicate that some children in Indonesia have not received the vaccination, and the HPV vaccine program has not touched men. The poor knowledge has been reflected in poor vaccination uptake, as only 17.3% have already received at least one dose of vaccination at least one dose in our study. However, 85% of participants were ready to get vaccinated. Prior studies demonstrated HPV vaccination uptake ranging from 70% to 100%, higher in well-informed people well-informed about the vaccine [92,93].

It is essential to identify influenced factors regarding vaccinations uptakes as a basis of policy to improve vaccination rates. Our multivariate logistic regression demonstrated that a female mother's education, salary per month, and prior knowledge become a solid predictor for vaccinating at least one dose. Salary has become an essential key to getting vaccination since the current market prices of HPV vaccines in Indonesia are considerably expensive for the general public, especially for people with a low socioeconomic status. Indonesia is now classified as a low-middle income country with a GDP per capita of 3,869.59 USD compared to other neighboring countries: Singapore (59,797.75 USD), Malaysia (10,401.79), and Thailand (7,189.04), or with countries where CC programs are well-established: US (63,543.58

USD), UK (40,284.64 USD), and Australia (51,812.15 USD) [94]. Moreover, it is not included in the national program and is not covered by medical insurance in Indonesia. These made it challenging to improve vaccine coverage, especially among males who perceived that the HPV vaccine was unsuited for them [95]. Indonesia market offers HPV vaccine with 52.08–90.27 USD per dose injection, similar with Phillippines (54 USD per dose) [96], cheaper compared to US (150–190 USD per dose) [97,98], but higher than Brazil (12.83 USD per dose) [98]. A study showed that HPV vaccination is not cost-effective when the price is above 50 USD per dose, especially for people living in rural areas [99]. In addition, Indonesia's self-financing of HPV vaccination policy demands a massive allocation since the country has to supply about five million (two doses x 2.5 million 10-year girls) doses annually [100]. Furthermore, the vaccination cost must consider the vaccine price, shipping, insurance, and handling fees (i.e., customs clearance, warehouse storage, transportation, and labeling), creating more costly [100].

We found that the intention and readiness to get vaccinated rely on sex, prior knowledge, and attitudes. In adolescents, parents also perform an essential role in vaccination. The parents' intentions to vaccinate their children were significantly influenced by the demand for further information about HPV, CC, and HPV vaccine [101]. About 85.6% of participants want more HPV, CC, and vaccination information. This result reflects that prior knowledge of HPV, CC, and the vaccine can change participants' eagerness to acquire HPV vaccine similar to previous research [69]. This phenomenon highlights the necessity to increase parents' KAP of HPV vaccination. The low level of vaccination is also possibly due to worries about side effects.

In Table 6, several factors arise from KAP aspects contributing to vaccinating one dose of HPV vaccination, including accessibility, dose acknowledgment, willingness, and not being worried about side effects. Meanwhile, fearing injections are not a barrier to getting vaccinated. Subsequently, high perceived protective effect, dangerous perspective on HPV and CC, enthusiasm, being fearless of HPV, and intention to seek more information about HPV become factors to be ready to get vaccinated similarly to previous findings [102]. Intention and readiness to get vaccinated against HPV were significantly associated with knowledge and attitudes [103]. Providing literacy about CC and HPV-related disease is essential to enhance vaccine uptake [104]. Proper education should be addressed to the male population due to their risk of being carriers and their role as family decision-makers. The acceptance rates are pretty high after proper socialization [105].

HPV infection rates in males have been reported to range from 1.3% to 72.9%, depending on the sampling and processing method and the anatomic sites taken [106]. Males should be concerned about the beneficial effects of HPV vaccination in preventing prostate and penile cancer. Given its anatomic proximity to anogenital and urinary sites, many studies investigated the association between HPV-16 and 18 infections and prostate cancer (PC), particularly among Mediterranean and Asian populations [107–114]. Yin et al. [114] found the pooled OR of HPV-associated PC risk was 2.27 (95% CI: 1.40–3.69). Globally, overall pooled prevalence of HPV in PC cases was 19% (CI 10.0%-29.0%) [115,116]. In 2020, the estimated PC incidence and mortality related to HPV in Indonesia were 9.85 and 3.53 per 100,000, becoming the seventh-highest death among males with HPV [10]. High-risk HPV subtype DNA was significantly higher in PCs (21.6%) as compared to normal and benign prostate controls (6.7%) [117]. They have access, probably through sexual transmission directly via the urethra, to immortalize and transform normal prostate cells into malignant cells by entangling large T antigen or E6/E7 proteins of HPV and meddling with the interferon signaling pathway [117–119]. Additionally, HPVs can be transmitted via circulating extracellular vesicles and blood [117]. High-risk HPVs are also associated with inflammatory prostatitis, leading to benign prostate hyperplasia and later PC. HPV infections may also initiate prostate oncogenesis via a

mutational mechanism involving an enzyme called APOBEC and collaborating with other pathogens in prostate oncogenesis [117]. There are differences between HPV-associated CC and PC characteristics. The HPV viral load is extremely low in PC as compared to CC. PC primarily involves glandular epithelial cells compared to squamous epithelial cells in CC [117]. In this decade, the increase of PCs has been reported in developing countries, possibly due to rapid urbanization, changing lifestyles, and a high rate of free unprotected sex, all significant risk factors for male acquisition of HPV infection (up to 40%) [106,117,120].

Besides in PC, multiple genotypes of HPV DNA can also be detected in penile and testicular cancer [121]. Around 40% of all penile tumors are attributable to HPV infections [122]. Estimated penile cancer incidence and mortality related to HPV in Indonesia during 2020 were 0.74 and 0.25 per 100,000, and the corresponding number for testicular cancer was 1.09 and 0.31 per 100,000 [10]. Although it is unknown how long HPV vaccines administered to young males can remain effective [117], the wide use of quadrivalent HPV vaccine has been recommended for young men to prevent genital warts and penile cancer.

Proper knowledge and a positive attitude towards the CC prevention program, including vaccination and screening, are the most critical factors for accepting these services in the community [103,123–125]. Besides, we observed that the respondents' demographic profiles significantly influenced the level of KAP. The sex, education level, age, employment status, income, and parent's education level, are the main factors that should be addressed to optimize the effectiveness of CC prevention in urban communities.

## Strength and limitations

We acknowledge several limitations in this study. The first constraint arises from the inherent study design, a cross-sectional study making it difficult to directly present the variables' causal relationship. Due to the lockdown situation, we had to rely on online surveying, which might be a second boundary. Online surveying is confined to only being completed by a literate person and able to comprehend the questions listed. This potentially caused unanswered questions, which has been anticipated by a system to make the questions unskippable. The third issue might be addressing the snowball sampling technique. It is a robust, wide-scope, highly accurate, suitable in limited resources, and cost-effective means to systematically capture valuable data from a broader range of audiences [126]. Nonetheless, we are aware of some limitations of this technique. Compared to the random sampling method, this technique might lessen the sample's representativeness and preclude generalizations to all urban citizens or the entire population. The snowball technique has little control over the sampling method and removes the researcher from the center of the sampling process. It might trigger sampling bias because respondents tend to forward the survey to peers with similar traits and characteristics. However, we anticipated this obstacle by conducting an exponential non-discriminative snowball sampling with a multi-referral strategy which allows us to have a broad characteristics sample [127]. There is no intention from the researcher to choose the sample subjectively since we distributed the questionnaire link randomly through social media [128]. The fourth limitation comes from this study being conducted only in Jakarta, a pioneer city of CC program implementation. However, our sample remains valuable and significant in picturing the Indonesian citizens due to Jakarta being a popular city to transmigrate for people from all over Indonesia. The dominant age group causes the fifth concern in our research, mostly being 11–30 years old made the results of this research might not be generalized to all ages. Moreover, as the sixth issue, our respondents were primarily females. They are more likely to be interested in this topic. They are also more engaged in online activities characterized by different communication styles than male subjects [127].

Despite those limitations, our findings are consistent with the outcomes of several studies with similar study methods [129,130]. Due to scarce data in this field, our work can contribute to the evidence of an association between KAP regarding HPV infection, CC, and HPV vaccination. Our sample from inhabitants of Jakarta pictured various backgrounds socioeconomic strata and an analysis of their influences on a proper understanding of CC and HPV infection prevention-related issues. Hopefully, this will be a basis for better CC prevention in Indonesia.

## Conclusions

The KAP regarding HPV infection, CC, and HPV vaccination are inter-associated with successful CC and HPV infection prevention programs. The knowledge level regarding these issues among Jakarta's urban community was poor in each aspect but moderate overall. Still, their attitude was positive overall and for each aspect; meanwhile, the practice was not satisfying. Generally, females had a better understanding of HPV-related issues. An alarming finding of our study was that HPV vaccination uptake for at least one dose was still low, although readiness to get vaccinated was good. There is a critical influence of socio-demographic characteristics to the KAP level towards HPV, CC, and HPV vaccination.

We recommend that all Indonesians aged 11–30 years acquire quadrivalent or nonavalent HPV vaccination; if not available, they can still use the bivalent vaccine. People with age <15 years old can still use two-dose vaccination and above it should be vaccinated with three doses. Susceptible people ≥15 years old, regardless of their sexual activity and infection status, may still need catch-up vaccination with three doses of HPV vaccine due to being beneficial to immunize against HPV serotypes that had not infected the person. This age group is also highly recommended to be tested for HPV DNA to get an early diagnosis. An alternative is for women in this age group to take IVA tests with or without following Papanicolau smear to screen for CC status.

Our study revealed an urgent need to enhance primary healthcare and government participation to increase KAP regarding HPV infection, CC, and the HPV vaccine for women and men. As a special note, a healthcare professional should maximally empower mothers since their knowledge is critical in influencing vaccination uptake for their children. Additionally, a framework of educational intervention among the youth through online media promoting CC and HPV-related disease prevention should be established. The Indonesian government should conduct a national vaccination program, ideally for free, and formulate a pricing policy related to an affordable retail price of HPV vaccines in the market because economic status is a significant barrier to vaccination uptake.

## Supporting information

**S1 Checklist. STROBE checklist.**
(PDF)

**S1 Table. Knowledge questions and responses from 400 respondents regarding HPV infection, CC, and HPV vaccination.**
(PDF)

**S2 Table. Attitude questions and responses from 400 respondents regarding HPV infection, CC, and HPV vaccination.**
(PDF)

**S3 Table. Practice questions and responses from 400 respondents regarding HPV infection, CC, and HPV vaccination.**
(PDF)

**S1 File. Questionnaires and scoring system.**
(PDF)

## Acknowledgments

### Ethics statement

The study protocols were approved by the Ethics Committee of the Faculty of Medicine, University of Indonesia, and Dr. Cipto Mangunkusumo Hospital with protocol number KET-237/ UN2.F1/ETIK/PPM.00.02/2020.

## Author Contributions

**Conceptualization:** Hariyono Winarto, Muhammad Habiburrahman, Maya Dorothea, Andrew Wijaya, Kartiwa Hadi Nuryanto.

**Data curation:** Hariyono Winarto, Muhammad Habiburrahman, Maya Dorothea, Andrew Wijaya.

**Formal analysis:** Hariyono Winarto, Muhammad Habiburrahman.

**Funding acquisition:** Hariyono Winarto.

**Investigation:** Hariyono Winarto, Muhammad Habiburrahman, Maya Dorothea, Andrew Wijaya.

**Methodology:** Hariyono Winarto, Muhammad Habiburrahman, Kartiwa Hadi Nuryanto.

**Project administration:** Muhammad Habiburrahman, Maya Dorothea, Andrew Wijaya.

**Resources:** Hariyono Winarto, Muhammad Habiburrahman, Maya Dorothea, Andrew Wijaya.

**Software:** Muhammad Habiburrahman.

**Supervision:** Hariyono Winarto, Kartiwa Hadi Nuryanto, Fitriyadi Kusuma, Tofan Widya Utami, Tricia Dewi Anggraeni.

**Validation:** Hariyono Winarto, Kartiwa Hadi Nuryanto, Fitriyadi Kusuma, Tofan Widya Utami, Tricia Dewi Anggraeni.

**Visualization:** Muhammad Habiburrahman.

**Writing – original draft:** Hariyono Winarto, Muhammad Habiburrahman.

**Writing – review & editing:** Hariyono Winarto, Muhammad Habiburrahman, Kartiwa Hadi Nuryanto, Fitriyadi Kusuma, Tofan Widya Utami, Tricia Dewi Anggraeni.

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
