## [Decision Letter · Decision Letter 0]

10 Jan 2022

PONE-D-21-35609Knowledge, attitudes, and practices among Indonesian urban communities regarding HPV infection, cervical cancer, and HPV vaccinationPLOS ONE

Dear Dr. Winarto,

Thank you for submitting your manuscript to PLOS ONE. After careful consideration, we feel that it has merit but does not fully meet PLOS ONE’s publication criteria as it currently stands. Therefore, we invite you to submit a revised version of the manuscript that addresses the points raised during the review process. Please aim to address the study shortcomings regarding the survey methodology raised by all reviewers especially about the inclusion/exclusion criteria, implementation/response collection and representativeness.Try to address the language issues and if possible, shorten the text or move parts to the supplemental material to make it more concise

We look forward to receiving your revised manuscript.

Kind regards,

Ivan Sabol

Academic Editor

PLOS ONE

Journal Requirements:

Additional Editor Comments:

The reference 20 at pate 7 L171 appears to be irrelevant for the study performed. However, additional information on the study design might be helpful, ie, how potential bias was avoided or corrected for. While not an expert on survey design i am aware of several biases in online questionnaires myself and would inquire whether particular steps were taken to minimize the bias

Please see the following links for some of the problems https://www.d8aspring.com/blog/4-types-of-biases-in-online-surveys-and-how-to-address-them

The Importance of Selection Bias in Internet Surveys. https://www.scirp.org/journal/paperinformation.aspx?paperid=67313

Participation Bias Assessment in Three High-Impact Journals. https://journals.sagepub.com/doi/full/10.1177/2158244013511260

The authors only state P26 L563 "Similar to many survey-based analyses, our research is prone to a response bias" which seems inadequate considering the length of the manuscript

While it is interesting to note what factors were associated with positive attitudes, it might be interesting to look at the problem other way around. The authors note that P25 L532 “It is essential to identify influenced factors regarding vaccinations uptakes as a basis of policy to improve vaccination rates”. Thus it might be important to find factors most associated with vaccine hesitancy (or lack of knowledge or poor attitudes) and focus policy towards those groups? The current study well describes people not needing policy attention?

The manuscript would benefit from some shortening as well as some language revisions

Reviewers' comments:

Reviewer's Responses to Questions

**Comments to the Author**

1. Is the manuscript technically sound, and do the data support the conclusions?

Reviewer #1: Yes

Reviewer #2: Partly

Reviewer #3: Yes

2. Has the statistical analysis been performed appropriately and rigorously? 

Reviewer #1: I Don't Know

Reviewer #2: Yes

Reviewer #3: No

3. Have the authors made all data underlying the findings in their manuscript fully available?

Reviewer #1: Yes

Reviewer #2: Yes

Reviewer #3: Yes

4. Is the manuscript presented in an intelligible fashion and written in standard English?

Reviewer #1: No

Reviewer #2: Yes

Reviewer #3: Yes

5. Review Comments to the Author

Reviewer #1: Winarto et al “Knowledge, attitudes, and practices among Indonesian urban communities regarding HPV infection, cervical cancer, and HPV vaccination”

Thankyou for asking me to review this manuscript.

Because of the high prevalence of cervical cancer in Indonesia, this is an important topic for both Indonesian women and men.

The concepts included in the study are excellent.

Of special interest to Western readers is the demonstration by this Indonesian group of the use of almost cost free social media to conduct an important study.

As shown in this study, HPV vaccination rates are low for both women and men (particularly men).

The results of the study indicate that an education campaign is required to increase the HPV vaccination rates.

Congratulations to the authors for their interest in this important topic and for their ability to conduct a study without cost. Well done!

In my view, after revision, the study should be published.

Limitations and suggestions

The following comments and suggestions are intended to be helpful. They are not criticisms. They mainly concern ways in which to improve the presentation of the manuscript.

1. The manuscript needs a review to improve the English writing.

2. There is an excessive use of acronyms (particularly in the abstract). These are distracting for the reader. There is advantage in using repeated full spelling of unusual phrases such as “… knowledge, attitudes, and practices (KAP)…”.

3. The methods used to select the participants in the surveys is not sound from a statistical basis. The participants are far from representative of the Indonesian urban population. For this study this is not important but the limitations need to be recognised and included in the methods.

4. Costs of HPV vaccines. I am not well informed but I understand that one reason for the low HPV vaccination rates in Indonesia is the cost to the recipient and their families. Income levels are low at $350 US dollars per month. Western readers are accustomed to much higher incomes This payment should be presented as per month each time it is included in the manuscript.

5. The manuscript is much too detailed. It is hard work reading the detail. I suggest much of the detailed data could be in Supplementary Tables. In addition much detailed data could be excluded from the narrative.

6. I suggest that the authors include a comparison of the Indonesian guidelines for HPV vaccination with those of the United States Center for Disease Control. In the US HPV vaccination is recommended for young teenagers. The US guide also recommend no HPV vaccination for people over the age of 26 (because of the likelihood of prior HPV infections). The situation may be very different in Indonesia. Sexual behaviour (the most common means of HPV transmission) is presumably different.

This reviewer is not from the US but many readers of this Journal are American.

7. I suggest the authors include the risk of HPVs in male prostate cancer in their discussions section. This may alter the interest in HPV vaccination by male Indonesians. (see recent reviews in this Journal). Also women catch HPV infections from men. Based on this study, the HPV vaccinations are much lower in males than in females

Reviewer #2: Abstract:

- the statement "No study explored Indonesian understanding of cervical cancer (CC) and the human

112 papillomavirus (HPV) vaccine"  It would be wiser if replaced with "Only few studies ...", knowing that there have been other studies conducted in Indonesia previously.

- It should be mentioned that the 400 respondents were recruited from cities in Jakarta Province, one of 34 provinces in Indonesia. Hence, it would be fair for the readers to consider the representativenes of the subjects compared to the whole urban communities of Indonesian population.

Methods:

- explain more detail about what form of questionnaire and platform used in the only survey

- why the survey process took too long time for collecting 400 subjects? It is mentioned from March 2020 to August 2021.  It needs to be explained because the time gap might influence the KAP due to enviromental changing in all aspect.

- what is the inclusion and exclusion criteria of the subjects?

Conclusion:

- It is concluded that the knowledge was poor. What was the cutting score apllied to categorized the knowledge level?

Reviewer #3: In general, the manuscript is interesting and provide evidence for enhancing effort of cervical cancer prevention, but need major revision, as follows:

1. Title

Title should in line with the study population which is Jakarta citizens, not all urban communities in Indonesia. We suggest you to revise it.

2. Abstract

a. Background: Revise the word “no study explored…”. There are many studies about HPV and cervical cancer. It’s better if you write there is limitation of study about it, but you should provide the studies in the background

b. Aim: what is the aim of the study? Association of KAP towards HPV infection, CC, and HPV vaccine? It’s different with the result that is association of socio-demographic characteristics with KAP in terms of HPV infection, CC, and HPV vaccination. Make it clear.

c. Method: add who the respondent are (Jakarta), sampling technique, how to get the respondent and sample size, how to collect data and when you did

d. Result: Make it simple as aim of the study, not all result. We suggest to delete median, just straight to say high or poor (KAP)

e. Conclusion: make it more specific related to the aim of the study.

3. Background

Explain why you choose urban communities. Cervical cancer is a disease not only in urban, but also in rural communities. Provide evidence, such as comparison incidence or prevalence in urban and rural areas, and related studies of KAP and cervical cancer. Again, word “ no study explored…” should be revised

4. Method

a. Describe how to spread the link/web in social media to invite the respondents.

b. Explain how sample size is counted, there are several independent and dependent variables to be considered.

c. Add information of time of data collection (is it more than a year from March 2020 – August 2021?). Is there any inclusion criteria/restriction of age? How a teenager of age of 11 years become respondent?

d. In assessment tools, we suggest to move contributory from authors to specific section, not in method.

e. In statistical analysis, add information of normality test, due to use of Spearman Correlation test. Add steps of multivariate logistic regression analysis, how the modeling constructed and how to get the final model (do you put the final model in the result?)

5. Result

a. Table 2 – 4 is not needed, its only step to get table 4. We suggest deleting table 2-4.

b. Table 6-9: are they the final model? You should give explanation. Put only the significant variables in the final model. make table 9 simpler.

6. Discussion

a. The age of respondents majority in age of 11-30 (80%), the must a specific recommendation in this age

b. Add limitation of the study in terms of sampling technique (snowball), that can not fully represent the population of Jakarta citizens/urban communities. The limitation of characteristic especially age, so result of the study may not be generalized to all ages.

7. Conclusion

Elaborate the most interesting findings related to the aim of the study. We suggest not to use term “domain” that is not easy to understand.

6. PLOS authors have the option to publish the peer review history of their article (what does this mean?). If published, this will include your full peer review and any attached files.

Reviewer #1: **Yes: **James Lawson

Reviewer #2: No

Reviewer #3: No

---

## [Author Response · Author response to Decision Letter 0]

24 Feb 2022

Dear Editor and Reviewers of PLOS One

The authors truly respect the time and effort Editor and Reviewers dedicated to providing constructive and insightful feedback on our manuscript. We appreciate the valuable input which allows us to revise our manuscript titled "Knowledge, Attitudes, and Practices Among Indonesian Urban Communities Regarding HPV Infection, Cervical cancer, and HPV Vaccination" (Manuscript ID = PONE-D-21-35609) for publication in PLOS ONE.

These are our comments for editor and reviewers:

#Editor: Thank you for taking the time to provide your feedback regarding the substance of our paper, language and concise issue, supporting information guidelines, reference, figure conversion following PACE digital diagnostic tool, and constructing valuable recommendations from the patient perspective to empower theirself without only depending to government. In addition, we would say thank you for your kindly remember regarding providing repository information. We have made significant changes within the manuscript, incorporating suggestions from reviewers and editors. Regarding your notes, we tried to address the study shortcomings regarding the survey methodology raised by all reviewers, especially about the inclusion/exclusion criteria, implementation/response collection, and representation. We also attempted to fix the language issues and shorten the text or move parts into supplemental material to make it more concise. Details of the changes we made can be seen through the Track Changes feature. We believe that all the comments from reviewers and editors help us improve our manuscript and are more valuable to the reader. All your comments and feedback will be discussed in the "response to reviewers" file. 

#Reviewer Number 1: Thank you for your generous comment and suggestions to improve our paper. We appreciate the details you shared about areas we can improve upon—this insight will help us significantly improve our paper. Hopefully, this article could enlighten readers about the HPV vaccination atmosphere in Indonesia among men and women. We admire your valuable input that this paper will be valuable in both low-to-middle income and developed countries. Living in a developing country is struggling; thus, all the research was conducted efficiently and effectively to minimize cost. Although it has several limitations arising from several aspects, we hope it still serves the best idea and research interests for readers worldwide involved in this topic. We have incorporated your suggestions into our revision. All your comments and feedback will be discussed in the "response to reviewers" file.

#Reviewer Number 2: We appreciate the time you have committed to providing us with detailed and constructive feedback. We have incorporated your suggestions into our revision. All your comments and feedback will be discussed in the file "response to reviewers."

#Reviewer Number 3: Thank you so much for your prompt and detailed feedback. We are very grateful for the time you took out of your day to write us detailed comments. We have outlined our responses and explanations in the table. We have incorporated your suggestions into our revision. All your comments and feedback will be discussed in the "response to reviewers" file.

Because there are many revisions in our manuscript, which resulted in quite a lot of changes in the reference citation arrangement and improvement in grammar, hence, in preventing missed data, we ask the reviewer and editor to use this latest version of our Word document for the following publication process. We also have proofread for any grammatical and wording issues in the revision process. Please use the Track Changes feature inherent to your word processing software to view revised sentences and phrases.

We hope that this revised manuscript will meet your requirements and be granted the opportunity to be published in PLOS ONE.

Thank you

Sincerely,

Hariyono Winarto, MD, Gynaecologist-Oncologist, PhD

Division of Gynecology Oncology, Department of Obstetrics and Gynecology, Faculty of Medicine, Universitas Indonesia and Dr. Cipto Mangunkusumo National Referral Hospital, Jakarta, Indonesia

---

## [Editor Report · Decision Letter 1]

15 Mar 2022

Knowledge, attitudes, and practices among Indonesian urban communities regarding HPV infection, cervical cancer, and HPV vaccination

PONE-D-21-35609R1

Dear Dr. Winarto,

We’re pleased to inform you that your manuscript has been judged scientifically suitable for publication and will be formally accepted for publication once it meets all outstanding technical requirements.

Kind regards,

Ivan Sabol

Academic Editor

PLOS ONE

Additional Editor Comments (optional):

All comments were addressed
---

## [Editor Report · Acceptance letter]

4 May 2022

PONE-D-21-35609R1 

Knowledge, attitudes, and practices among Indonesian urban communities regarding HPV infection, cervical cancer, and HPV vaccination 

Dear Dr. Winarto:

I'm pleased to inform you that your manuscript has been deemed suitable for publication in PLOS ONE. Congratulations! Your manuscript is now with our production department. 

Kind regards, 

on behalf of

Dr. Ivan Sabol 

Academic Editor

PLOS ONE